# Use of the Serum *Wisteria floribunda* Agglutinin-Positive Mac2 Binding Protein as a Marker of Gastroesophageal Varices and Liver-Related Events in Chronic Hepatitis C Patients

**DOI:** 10.3390/diagnostics10030173

**Published:** 2020-03-22

**Authors:** Tsuguru Hayashi, Nobuharu Tamaki, Masayuki Kurosaki, Wan Wang, Mao Okada, Mayu Higuchi, Kenta Takaura, Hitomi Takada, Yutaka Yasui, Kaoru Tsuchiya, Hiroyuki Nakanishi, Jun Itakura, Masaru Harada, Namiki Izumi

**Affiliations:** 1Department of Gastroenterology and Hepatology, Musashino Red Cross Hospital, Tokyo 180-8610, Japan; haya033185@yahoo.co.jp (T.H.); tamaki@musashino.jrc.or.jp (N.T.); kurosaki@musashino.jrc.or.jp (M.K.); w.ou@musashino.jrc.or.jp (W.W.); m.okada@musashino.jrc.or.jp (M.O.); mayu621201@yahoo.co.jp (M.H.); tuf029@gmail.com (K.T.); takadahi0107@gmail.com (H.T.); yutakay@musashino.jrc.or.jp (Y.Y.); tsuchiya@musashino.jrc.or.jp (K.T.); nakanisi@musashino.jrc.or.jp (H.N.); jitakura@musashino.jrc.or.jp (J.I.); 2Third Department of Internal Medicine, School of Medicine, University of Occupational and Environmental Health, Kitakyushu 807-8555, Japan; msrharada@med.uoeh-u.ac.jp

**Keywords:** WFA^+^–M2BP, chronic hepatitis C, gastroesophageal varices

## Abstract

Background: A test to narrow down patients who require esophagogastroduodenoscopy (EGD) with a high probability of having gastroesophageal varices (GEV) and a high-risk of liver-related events is an unmet need. Methods: The measurement of serum fibrosis markers and EGD was performed in 166 consecutive chronic hepatitis C patients. The correlation between the grades of GEV and fibrosis markers and the subsequent occurrence of liver-related and fibrosis markers were examined. Results: *Wisteria floribunda* agglutinin-positive human Mac-2 binding protein (WFA^+^–M2BP) levels increased according to the grade of GEV (3.4 (0.2–18.6) for no GEV, 7.9 (1.8–20.0) for small GEV, and 11.4 (4.0–20.0) for large GEV; *p* < 0.001). The diagnostic accuracy of the WFA^+^–M2BP was superior compared to other serum fibrosis markers, and WFA^+^–M2BP was an independent predictor of GEV in the multivariate analysis. Furthermore, the cumulative incidence of liver-related events at one year was 2.3% in patients with WFA^+^–M2BP levels ≤ 7.0 and 37.5% in patients with WFA^+^–M2BP levels > 7.0 (*p* < 0.001). WFA^+^–M2BP > 7.0 was a significant predictive factor for liver-related events (Hazard ratio 6.7, *p* = 0.004) independent of Child–Pughclass. Conclusions: WFA^+^–M2BP could be used to estimate the presence and grade of GEV and is linked to liver-related events in chronic hepatitis C patients.

## 1. Introduction

Portal hypertension is the most serious complication of chronic liver disease, and the gastroesophageal varices (GEV) caused by portal hypertension are present in 50% of liver cirrhosis patients [1]. Among these, large GEV are prone to bleeding at a rate of 5–15% a year [1]. GEV remains a serious and sometimes fatal complication despite recent advancements in endoscopic therapy [2,3]. Small GEV can develop into large GEV at a rate of 8% a year [1]. Therefore, it is recommended that patients undergo esophagogastroduodenoscopy (EGD) once every 1–2 years if they present small GEV and about once every 2–3 years if they present no GEV [1]. However, it may be uneconomical to perform EGD on patients who have a low risk of GEV. Moreover, EGD is not readily available in areas with limited resources. A test to narrow the patients who require EGD and have a high probability of suffering from GEV is an unmet need.

To date, blood tests, such as platelet counts, the FIB-4 index, the aspartate transaminase to platelet ratio index (APRI) and FibroTest, and imaging tests, such as acoustic radiation force impulse (ARFI) imaging and transient elastography (TE), have been used to predict the degree of fibrosis or the presence of GEV [4,5,6,7,8] However, blood tests for platelet counts, the FIB-4 index, APRI, and other parameters have low sensitivity and specificity for estimating GEV [9]. Furthermore, TE and ARFI have several drawbacks, such as reduced accuracy in obese patients or those with ascites, and are influenced by the inflammation of the liver [10]. None of these indices are sufficient for estimating GEV. An accurate and simple marker for estimating GEV is, therefore, awaited.

The *Wisteria floribunda* agglutinin-positive human Mac-2 binding protein (WFA^+^–M2BP) was developed as a non-invasive serum marker for predicting fibrosis in chronic hepatitis C [11] and has been subsequently used as an accurate and simple tool for evaluating liver fibrosis [12,13,14,15]. Because GEV is proportional to the degree of liver fibrosis [16], we hypothesize that serum WFA+–M2BP, which increases as fibrosis progresses, has potential to estimate GEV.

In this study, we investigate the utility of serum WFA+–M2BP measurement in the estimation of the presence and the grade of GEV in patients with chronic hepatitis C by comparing WFA+–M2BP with other non-invasive markers (FIB-4 index, platelet count, APRI, and alanine aminotransferase ratio (AAR)). We also prospectively examined the occurrence of liver-related events after the determination of serum WFA+–M2BP levels.

## 2. Methods

### 2.1. Patients

A total of 166 consecutive chronic hepatitis C patients who underwent EGD between April 2015 and March 2016 were enrolled. Patients at all fibrosis stages, not just those with cirrhosis, were enrolled in the study. Hepatitis C virus (HCV) infection was confirmed by detecting HCV RNA by polymerase chain reaction (Rochie Molecular Diagnostics, Tokyo, Japan). Patients with a history of portal vein thrombosis, liver transplant, and co-infection with hepatitis B virus were excluded. WFA+–M2BP measurements and blood tests for aspartate transaminase (AST), alanine transaminase (ALT), platelet count, and other parameters were measured within one month before and after EGD. Written informed consent was obtained from each patient. The study protocol was approved (approval number:28089 4/Apr/2017) by the ethics review committees of Musashino Red Cross Hospital and conformed to the ethical guidelines of the Declaration of Helsinki.

### 2.2. Endoscopic Evaluation of GEV

EGD was performed by endoscopists at our hospital. The GEV grade was categorized into three stages (no GEV, small GEV, and large GEV) in accordance with the American Association for the Study of Liver Diseases guidelines.

### 2.3. WFA+–M2BP and other Serum Marker Measurements

Serum WFA+–M2BP was measured based on a lectin-Ab sandwich immunoassay using the fully automatic immunoanalyzer, HISCL2000i (Sysmex, Hyogo, Japan) [11]. The FIB-4 index [17], APRI [18], and AAR [19] were calculated according to previous established formulas.

### 2.4. Occurrence of Liver-Related Events

Liver-related events were defined as bleeding from the GEV, hepatic encephalopathy, ascites retention, and death. The occurrence of liver-related events was prospectively observed from the day of WFA+–M2BP measurement.

### 2.5. Statistical Analysis

Categorical data were compared using chi-squared and Fisher’s exact tests. Continuous variables were analyzed using the Mann–Whitney U-test. Receiver operating characteristic (ROC) curves and areas under the ROC curve (AUROCs) were used to evaluate the diagnostic accuracy of the tests, including the WFA+–M2BP, FIB-4 index, platelet count, AAR, and APRI tests. The cumulative incidence of liver-related events was determined by the Kaplan–Meier method, and the differences among groups were assessed using a log-rank test. The factors associated with the presence of GEV, the presence of a large GEV, and liver-related events were analyzed using the Cox-proportional hazard model. Correlated factors with *p*-values < 0.05 in the univariate analysis were used for a further multivariate analysis. The backward stepwise selection method was used for the multivariate analyses. The statistical analyses were performed using EZR (Saitama Medical Center, Jichi Medical University, Saitama, Japan) and a graphical user interface for R (The R Foundation for Statistical Computing, Vienna, Austria) [20].

## 3. Results

### 3.1. Patient Characteristics

Of the 166 patients with chronic hepatitis C, 87 had no GEV, 28 had small GEV, and 51 had large GEV. Upon comparing the patients with no GEV and those with the presence of GEV, the AST and prothrombin times were significantly higher in the patients with GEV. Furthermore, patients with the presence of GEV had a lower platelet count and albumin level and a worse Child–Pugh classification (Table 1). The median (first quartile–third quartile) follow-up period was 353 (253–478) days.

### 3.2. Serum WFA+–M2BP Levels and the Presence or Grade of GEV

The WFA+–M2BP level was 3.4 (range: 0.2–18.6) in patients with no GEV, which was significantly lower compared to the 7.9 (range: 1.8–20.0) in patients with a small GEV (*p* < 0.001, Figure 1) and 11.4 (range: 4.0–20.0) in patients with large GEV (*p* < 0.001). WFA^+^-M2BP levels were elevated in accordance with an increase in the grade of GEV. The optimal cut-off value of WFA+–M2BP for determining the presence of GEV was 6.0, with an AUROC of 0.90, a sensitivity of 78.5%, a specificity of 87.4%, a positive predictive value (PPV) of 84.9%, and a negative predictive value (NPV) of 81.7%. This performance was superior to that of the FIB-4 index, APRI, platelet count, and AAR (Table 2, Figure 2A). Furthermore, the optimal cut-off value of WFA+–M2BP for determining large GEV was 7.0, with an AUROC of 0.9, a sensitivity of 90.2%, a specificity of 80.9%, a PPV of 66.7%, and an NPV of 93.0%. This performance was also superior to that of the FIB-4 index, APRI, platelet count, and AAR (Table 2, Figure 2B).

### 3.3. Factors Associated with the Presence of GEV and Large GEV

In a univariate analysis, WFA+–M2BP was a significant factor associated with the presence of GEV (odds ratio (OR): 25.2, 95% confidence interval (CI): 11.0–57.7; *p* < 0.001; Table 3) and the presence of large GEV (OR: 38.9, 95%CI: 13.8–109; *p* < 0.001). A multivariate analysis demonstrated that WFA+–M2BP levels were an independent factor associated with the presence of GEV (OR: 30.7, 95%CI: 9.1–104.0; *p* < 0.001) and large GEV (OR: 28.4, 95%CI: 7.8–103.0; *p* < 0.001).

### 3.4. WFA+–M2BP Levels and Liver-Related Events

We prospectively examined the occurrence of liver-related events after stratification based on the serum WFA+–M2BP levels of 7.0 according to the cut-off value for large GEV. The cumulative incidence of liver-related events at one year was 2.3% in patients with WFA+–M2BP levels ≤7.0 and 37.5% in patients with WFA+–M2BP levels >7.0, indicating a significantly higher incidence of events in patients with WFA+–M2BP levels >7.0 (*p* < 0.001, Figure 3). Hepatic encephalopathy occurred in 4 patients, ascites retention in 12 patients, bleeding from GEV in 5 patients, spontaneous bacterial peritonitis in 2 patients, and death in 6 patients. WFA+–M2BP was significantly associated with liver-related events in the univariate analysis (HR: 14.8, 95%CI: 4.5–49; *p* < 0.001), and WFA+–M2BP levels were a significant predictive factor for liver-related events independent of Child–Pugh class in the multivariate analysis (HR: 6.7, 95%CI: 1.8–24; *p* = 0.004; Table 4).

## 4. Discussion

In this study, we investigate the association of serum WFA+–M2BP levels with GEV or liver-related events in patients with chronic hepatitis C. Serum WFA+–M2BP levels were significantly correlated with the presence of GEV and the grade of GEV and was the most accurate predictor of GEV among the various serum liver fibrosis markers. Furthermore, liver-related events occurred at a high rate in those with elevated WFA+–M2BP, suggesting that WFA+–M2BP could be a novel liver-related prognostic marker.

One finding of this study was the association between WFA+–M2BP and GEV. Some studies have shown the association between WFA+–M2BP and liver failure, hepatocellular carcinoma development, and its prognosis [21,22,23]. However, studies on the prediction of GEV are rare. Therefore, we believe our findings are important for clinical management. Liver fibrosis and GEV have a strong correlation. Many reports to date have described the utility of serum fibrosis markers or imaging techniques in estimating GEV [24,25,26,27,28]. Although serum markers, including the FIB-4 index, APRI, platelet count, and AAR, are simple, the diagnostic accuracy in estimating GEV is not sufficient [29]. Reports to date have described the same levels of diagnostic performance in estimating GEV as those found in our study. Meanwhile, imaging-based diagnoses, including TE and ARFI, present better diagnostic performance. However, the cost and time of these methods are drawbacks, and these modalities are not readily available. Furthermore, TE measurements are difficult to perform in obese or ascitic patients, and ARFI is influenced by elevated ALT, which makes the accurate assessment of fibrosis difficult [10]. In our results, the AUROC was 0.90 (sensitivity: 86.1%, specificity: 80.0%) to determine the presence of GEV and 0.90 (sensitivity: 78.1%, specificity: 90.9%) to determine the presence of large GEV. These results show significantly superior performance compared to other reports on imaging-based diagnoses, such as TE and ARFI.

In the Report of the Baveno VI Consensus, patients were screened for TE (>20 kPa) and platelet counts (<150,000/µL); narrowing down patients by screening is recommended for EGD [30]. The monitoring of both platelet count and TE is needed because TE produces many false-positive results [30]. Because the specificity of WFA+–M2BP in detecting large GEV is extremely high, and the measurement of WFA+–M2BP is simple, WFA+–M2BP could offer a better, more non-invasive, screening tool for detecting GEV. Of course, EGD is the best screening test for GEV. However, GED cannot be performed in all patients due to its invasiveness and economic drawbacks. Therefore, the measurement of WFA+–M2BP is useful to narrow down high-risk cases and reduce the number of unnecessary EGD screening tests. Appropriate EGD tests based on WFA+–M2BP screening and prophylactic therapy for high risk GEV may reduce the incidence of fatal events, such as the rupture of GEV.

Furthermore, we revealed in this study that WFA+–M2BP is also a predictor of liver-related events. The Child–Pugh classification is generally used to evaluate liver function when determining treatment strategies and predicting complications in liver cirrhosis patients. We found that the incidence of liver-related events increased significantly as WFA+–M2BP levels increased. WFA+–M2BP is associated with liver-related events independently of the Child–Pugh classification. In other words, while the Child–Pugh classification is widely used in the prognosis of liver cirrhosis and the prediction of complications, WFA+–M2BP could be a novel prognostic marker capable of predicting the prognosis of cirrhotic patients independently of the Child–Pugh classification. Many previous reports have discussed the diagnosis of liver fibrosis by WFA+–M2BP; however, our study discussed the usefulness of WFA+–M2BP not only in the diagnosis of liver fibrosis but also in stratifying the complications of patients with liver disease.

There are some limitations to this study, such as the study’s small sample size. Because the clinical status of the patients was not blinded to the endoscopists, this information may have affected the grading of GEV. Currently, most patients of chronic hepatitis C receive direct acting antiviral therapy and achieve sustained virologic responses. In this study, we investigated only patients with current HCV infections. In future research, we must investigate the time-course changes in GEV and WFA+–M2BP caused by antiviral therapy. Further, we only examined patients with chronic hepatitis C. The utility of WFA+–M2BP was proven in chronic hepatitis B, fatty liver disease, and other chronic liver diseases [31,32,33,34]. The cut-off value for stratifying the fibrosis stage differs depends on the etiology [35,36,37]. Therefore, it is necessary to study more cases with other etiologies in future studies.

In conclusion, WFA+–M2BP is a utility marker for the estimation of GEV and liver-related events in chronic hepatitis C patients and could be used to define patients who need EGD or prophylactic therapy for high risk GEV.

## Figures and Tables

**Figure 1 diagnostics-10-00173-f001:**
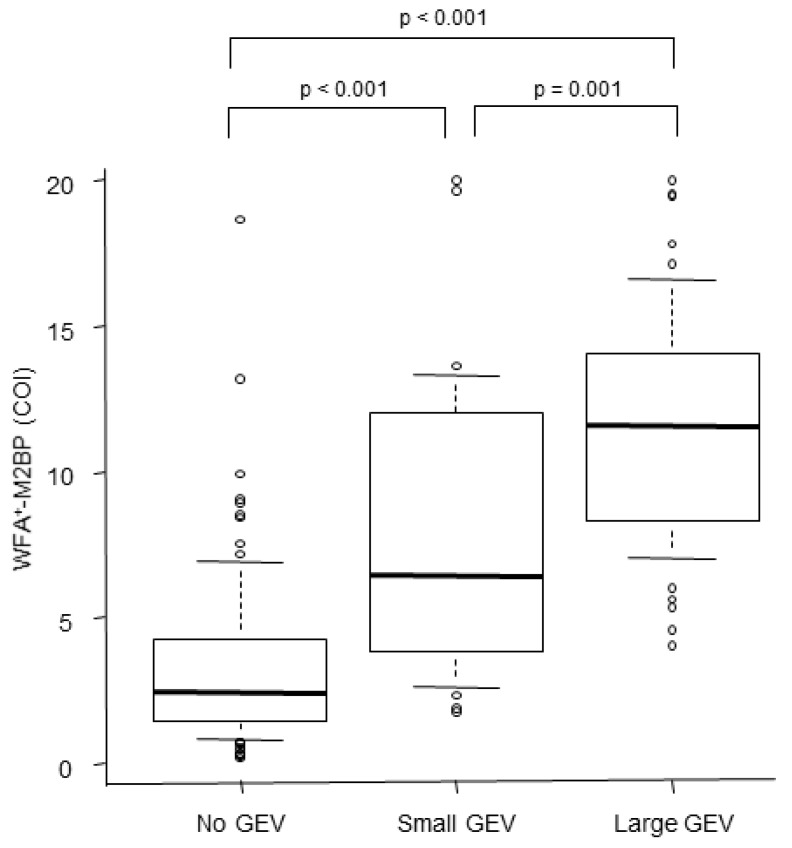
Correlation between the WFA+–M2BP level and GEV. A boxplot of WFA+–M2BP is shown for each grade of GEV. The boxplot represents the 25th to 75th percentiles and gives the interquartile range. The line through the box indicates the median value, and an error bar indicates the minimum and maximum nonextreme values.

**Figure 2 diagnostics-10-00173-f002:**
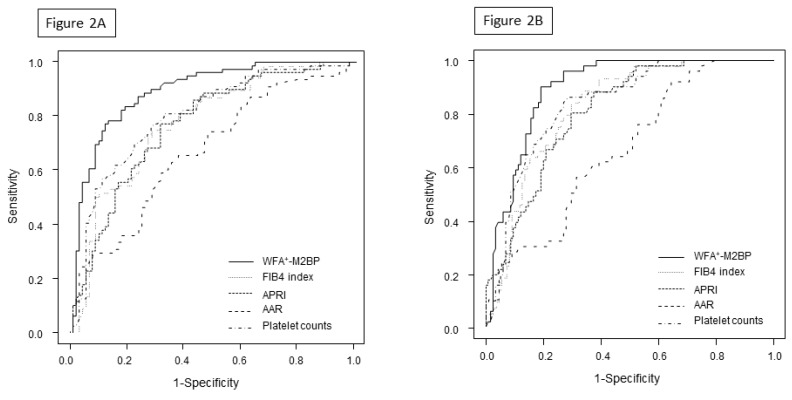
Receiver operating characteristic curves of WFA+–M2BP and other serum fibrosis markers. Various serum markers of fibrosis were compared to ensure the diagnostic accuracy of the presence of GEV (**A**) and the presence of large GEV (**B**).

**Figure 3 diagnostics-10-00173-f003:**
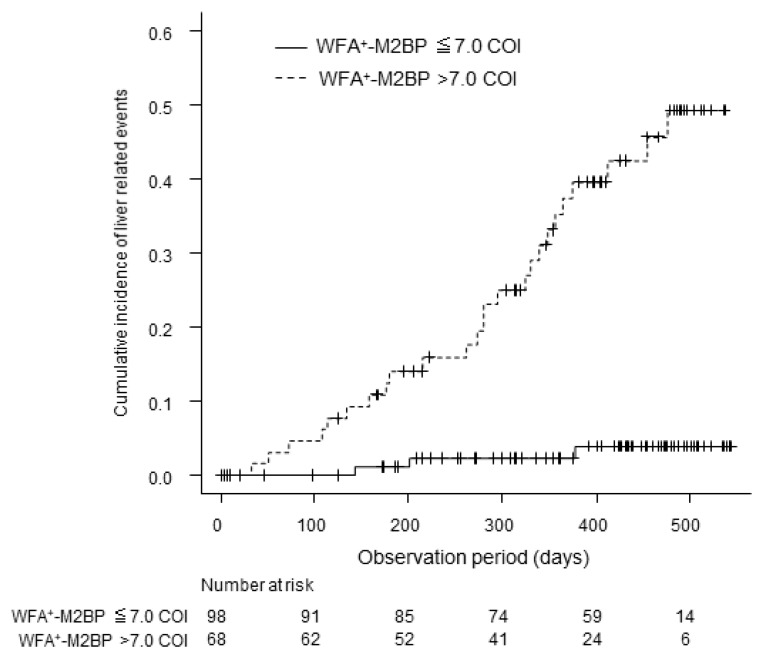
Cumulative incidence of liver related events. Patients were categorized into two groups with the cut-off value for WFA+–M2BP.

**Table 1 diagnostics-10-00173-t001:** Patient characteristics.

Factors	Category	No GEV	Present GEV	*p*-Value
Number		87	79	
Gender	male/female	37/50	44/35	0.09
Age (years)		72.5 ± 9.1	71.3 ± 10	0.42
AST (IU/L)		52.1 ± 34	71.6 ± 44	0.002
ALT (IU/L)		47.2 ± 45	48.9 ± 34	0.78
Albumin (g/dl)		3.86 ± 0.6	3.22 ± 0.5	<0.001
Platelet counts (×10^4^/μL)		16.2 ± 21	8.7 ± 4.0	0.003
Prothrombin Time (INR)		1.06 ± 0.14	1.16 ± 0.14	<0.001
Child-Pughclass	A/B/C	79/8/0	46/33/0	<0.001
History of HCC	No/Yes	60/27	35/44	0.001

AST, aspartate transaminase; ALT, alanine aminotransferase; HCC, hepatocellular carcinoma; INR, international normalized ratio; GEV, gastroesophageal varices.

**Table 2 diagnostics-10-00173-t002:** Diagnostic performance of WFA+–M2BP and other serum markers for detecting GEV.

Factors	AUROC	Cut-Off Value	Sensitivity	Specificity	PPV	NPV
Presence of GEV						
WFA+–M2BP	0.90	6.0	78.5	87.4	84.9	81.7
APRI	0.78	1.4	77.2	69.0	68.6	75.0
FIB-4 Index	0.81	6.0	78.5	72.4	72.0	76.2
Platelet counts	0.81	11.0	75.9	72.4	71.4	76.8
AAR	0.69	1.4	65.8	63.2	61.5	64.8
Presence of large GEV						
WFA+–M2BP	0.90	7.0	90.2	80.9	66.7	93.0
APRI	0.81	1.8	80.4	70.4	53.4	87.1
FIB-4 Index	0.84	6.4	82.4	72.2	56.2	89.2
Platelet counts	0.84	10.0	84.3	73.0	58.1	91.3
AAR	0.67	1.5	56.9	67.8	42.4	75.7

WFA+–M2BP, *Wisteria floribunda* agglutinin-positive human Mac-2 binding protein; PPV, positive predictive value; NPV, negative predictive value; GEV, gastroesophageal varices; AUROC: area under the receiver operating characteristics curve.

**Table 3 diagnostics-10-00173-t003:** Factors associated with presence of GEV and large GEV.

Presence of GEV	Univariate Analysis			Multivariate Analysis		
	Odds Ratio	95%CI	*p* Value	Odds Ratio	95%CI	*p* Value
WFA+–M2BP (>6.0 COI)	25.2	11.0–57.7	<0.001	30.7	9.1–104.0	<0.001
Child-Pughclass (B)	7.1	3.0–16.6	<0.001			
age (>70 years)	0.7	0.4–1.4	0.3			
gender (male)	1.7	0.9–3.1	0.09			
AST (>40 IU/L)	3.3	1.7–6.5	<0.001			
ALT (>40 IU/L)	2.1	1.1–3.8	0.02			
PT-INR (>1.1)	4.5	2.4–8.7	<0.001			
albumin (<3.5 g/dl)	5.2	2.7–10.2	<0.001			
Platelet counts (<10.0 × 10^4^/μL)	7.7	3.9–15.5	<0.001	4.4	1.7–11.2	0.002
History of HCC	2.8	1.5–5.3	0.001			
WFA^+^-M2BP (>7.0 COI)	38.9	13.8–109.0	<0.001	28.4	7.8–103.0	<0.001
Child-Pughclass (B)	4.4	2.1–9.3	<0.001			
age (>70 years)	0.4	0.2–0.9	0.02			
gender (male)	1.1	0.6–2.2	0.7			
AST (>40 IU/L)	4.6	2.0–10.6	<0.001			
ALT (>40 IU/L)	2	1.0–3.8	0.04			
PT-INR (>1.1)	6.4	3.1–13.5	<0.001			
albumin (<3.5 g/dl)	4.2	2.1–8.4	<0.001			
Platelet counts (<10.0 × 10^4^/μL)	12.6	5.5–29.0	<0.001	6	2.1–17.6	0.001
History of HCC	1.8	0.9–3.5	0.08			

WFA+–M2BP, Wisteria floribunda agglutinin-positive human Mac-2 binding protein; GEV, gastroesophageal varices; CI, confidence interval; AST, aspartate transaminase; ALT, alanine aminotransferase; HCC, hepatocellular carcinoma; INR, international normalized ratio.

**Table 4 diagnostics-10-00173-t004:** Multivariate analysis for liver related events.

Factors	Univariate Analysis			Multivariate Analysis		
	Odds Ratio	95%CI	*p* Value	Odds Ratio	95%CI	*p* Value
WFA+–M2BP (>7.0 COI)	14.8	4.5–49	<0.001	6.7	1.8–24	0.004
Child–Pughclass (B)	11.5	5.1–26	<0.001	5	2.0–12	<0.001
age (>70 years)	0.96	0.5–2.0	0.9			
gender (male)	2.3	1.04–4.9	0.04			
AST (>40 IU/L)	17.4	2.3–127	0.006			
ALT (>40 IU/L)	2.7	1.3–6.0	0.01			
Platelet counts (<10.0 × 10^4^/μL)	3.4	1.5–7.4	0.002			
History of HCC	3.6	1.6–8.0	0.001			

WFA+–M2BP, Wisteria floribunda agglutinin-positive human Mac-2 binding protein; AST, aspartate transaminase; ALT, alanine aminotransferase; HCC, hepatocellular carcinoma; INR, international normalized ratio.

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
