# Peer review of "Use of the Serum Wisteria floribunda Agglutinin-Positive Mac2 Binding Protein as a Marker of Gastroesophageal Varices and Liver-Related Events in Chronic Hepatitis C Patients"

_diagnostics, 2020, doi:10.3390/diagnostics10030173_

Round 1

Reviewer 1 Report

The authors evaluated the role of WFA+ -M2BP in predicting the presence/the grade of GEV and the occurrence of other liver-related events in patients with chronic hepatitis C.

The study is interesting and can provide some important information for clinical management.

The introduction is concise, provides sufficient background and include all relevant references.

The research design is appropriate, but the sample size is small and the study is monocentric.

There are some points which need to be better clarify:

  • The duration (median, IQ) of follow-up for the liver-related events;
  • Had all patient a diagnosis of cirrhosis? If yes, according which criteria;
  • Did the authors test normality of data distribution in order to analyse the continuous variables?
  • Are the endoscopists aware of patients clinical status (i.e. Child-Pugh score)? If yes, this point should be mentioned in the study limitation.

At line 177 there is an error in the platelet count which is:” >150,000/ml” NOT “>15,000/ml” according the Baveno VI Consensus.

The results are clearly presented and support the conclusions.

This paper should be accepted after minor revision.

Author Response

Reply to the reviewer

              Thank you for your kind review of our manuscript. Taking into account all the questions and criticism raised by the reviewers, we have now revised our manuscript. The changes made in the text are highlighted by red color text. Point-by-point reply to the reviewer’s comments is also attached. We believe that we have successfully answered to all the questions and criticisms, and that our manuscript is now improved and is suitable for the publication. We look forward to hearing from you.

Sincerely

Q: The duration (median, IQ) of follow-up for the liver-related events;

A: We added the median and 1-3IQ of follow-up in line104.

Q: Had all patient a diagnosis of cirrhosis? If yes, according which criteria;

A: Patients with all fibrosis stage, not just cirrhosis were included in the study. We added this statement in the method section (line64).

Q: Did the authors test normality of data distribution in order to analyse the continuous variables?

A: Since the continuous variables are not normally distributed, a nonparametric test was used.

Q: Are the endoscopists aware of patients clinical status (i.e. Child-Pugh score)? If yes, this point should be mentioned in the study limitation.

A: We added this point as the limitation of study (line207)

Q: At line 177 there is an error in the platelet count which is:” >150,000/ml” NOT “>15,000/ml” according the Baveno VI Consensus.

A: We corrected it (line185).

Reviewer 2 Report

In this study, the authors attempt to study the Wisteria floribunda agglutinin-positive human Mac-2 binding protein (WFA+ - M2BP) as a potential biomarker to narrow down patients requiring esophagogastroduodenoscopy (EGD) with a high probability of having gastroesophageal varices (GEV) as well as have high-risk of liver-related events. The authors measured serum fibrosis markers and EGD in 166 consecutive chronic hepatitis C patients.  After that authors examined the Correlation between the grades of GEV and fibrosis markers and subsequent occurrence of liver-related and fibrosis markers. It was observed that along with the grades of GEV, Wisteria floribunda agglutinin-positive human Mac-2 binding protein (WFA+- M2BP) levels also increased accordingly.  The authors observed and report that the diagnostic accuracy of WFA+-M2BP was superior as compare to other markers, and it was an independent predictor of GEV in multivariate analysis. Authors concluded that WFA+ -M2BP can be used as a biomarker to estimate the presence and grade of GEV, as well as it is linked with liver-related events in chronic hepatitis C patients.

Overall it is a good study.

Following are some suggestions/concerns;

  • The authors have used a relatively smaller data set (166 patients)
  • WFA+ -M2BP has been associated with various chronic liver diseases including fibrotic nonalcoholic steatohepatitis (NASH) and NASH cirrhosis, primary biliary cirrhosis (PBC), development of hepatocellular carcinoma besides hepatitis C patients, as per studies present in literature. And authors should provide a more detailed account of it. Itself, the manuscript is not more informative than other articles in the literature. The authors need to review in detail the published literature and identify the differences with the literature.
  • The authors have tested the association of this particular biomarker only in Hep C patients. They have mentioned it in the title, but inconsistently and also illogically in the manuscript. The authors should make it obvious in the introduction section. Both biomarkers and GEV can be present in different liver conditions too. Hence it is a bit confusing for the reader.
  • Data presentation needs to be improved.
  • The quality of English can be improved.

Author Response

Reply to the reviewer

              Thank you for your kind review of our manuscript. Taking into account all the questions and criticism raised by the reviewers, we have now revised our manuscript. The changes made in the text are highlighted by red color text. Point-by-point reply to the reviewer’s comments is also attached. We believe that we have successfully answered to all the questions and criticisms, and that our manuscript is now improved and is suitable for the publication. We look forward to hearing from you.

Sincerely

Q: The authors have used a relatively smaller data set (166 patients)

A: We added this point as the limitation of study (line207)

Q: WFA+ -M2BP has been associated with various chronic liver diseases including fibrotic nonalcoholic steatohepatitis (NASH) and NASH cirrhosis, primary biliary cirrhosis (PBC), development of hepatocellular carcinoma besides hepatitis C patients, as per studies present in literature. And authors should provide a more detailed account of it. Itself, the manuscript is not more informative than other articles in the literature. The authors need to review in detail the published literature and identify the differences with the literature.

A: Studies investigating association between GEV and WFA+ -M2BP have not been performed sufficiently and we believe our findings support clinical practice. We agree that further investigation is need in other etiology. These points were added in discussion section and limitation.(line169-172, 213-215) 

Q: The authors have tested the association of this particular biomarker only in Hep C patients. They have mentioned it in the title, but inconsistently and also illogically in the manuscript. The authors should make it obvious in the introduction section. Both biomarkers and GEV can be present in different liver conditions too. Hence it is a bit confusing for the reader.

A: To avoid confusing, we added the statement ‘patients with chronic hepatitis C’ in all section.

Q: Data presentation needs to be improved.

A: Sorry for not responding properly.

If you have any feedback, I will follow you.

Q: The quality of English can be improved.

A: The paper was submitted to MDPI English proofreading and proofread.

Round 2

Reviewer 2 Report

The authors seem to have addressed properly to the issues and concerns of the reviewers.